# The effect of generic market entry on antibiotic prescriptions in the United States

Cecilia Kållberg [1,2 ✉], Jemma Hudson[3], Hege Salvesen Blix[2,4], Christine Årdal [2], Eili Klein[5,6,7], Morten Lindbæk[1], Kevin Outterson [8,9], John-Arne Røttingen[1,2,11] & Ramanan Laxminarayan [5,10]

When patented, brand-name antibiotics lose market exclusivity, generics typically enter the market at lower prices, which may increase consumption of the drug. To examine the effect of generic market entry on antibiotic consumption in the United States, we conducted an interrupted time series analysis of the change in the number of prescriptions per month for antibiotics for which at least one generic entered the US market between 2000 and 2012. Data were acquired from the IQVIA Xponent database. Thirteen antibiotics were analyzed. Here, we show that one year after generic entry, the number of prescriptions increased for five antibiotics (5 to 406%)—aztreonam, cefpodoxime, ciprofloxacin, levofloxacin, ofloxacin —and decreased for one drug: cefdinir. These changes were sustained two years after. Cefprozil, cefuroxime axetil and clarithromycin had significant increases in trend, but no significant level changes. No consistent pattern for antibiotic use following generic entry in the United States was observed.

[1] Faculty of Medicine, Institute of Health and Society, University of Oslo, Oslo, Norway. [2] Norwegian Institute of Public Health, Oslo, Norway. [3] Health Services Research Unit, University of Aberdeen, Aberdeen, UK. [4] School of Pharmacy, University of Oslo, Oslo, Norway. [5] Center for Disease Dynamics, Economics and Policy, Washington, DC, USA. [6] Department of Emergency Medicine, Johns Hopkins School of Medicine, Baltimore, MD, USA. [7] Department of Epidemiology, Johns Hopkins Bloomberg School of Public Health, Baltimore, MD, USA. [8] School of Law, Boston University, Boston, MA, USA. [9] CARB-X, Boston, MA, USA. [10] Princeton University, Princeton, NJ, USA. [11] Present address: The Research Council of Norway, Lysaker, Norway. ✉email: ceciliakallberg@gmail.com

Antimicrobial resistance (AMR) is a global challenge that raises mortality from infectious diseases and increases healthcare costs[1,2]. Antibiotic use is a primary driver of AMR[3] and remains at a high level in the United States, despite efforts to improve prescribing practices and discourage inappropriate use[4,5]. As much as 30% of oral antibiotic use in the United States may be unnecessary[6]. Apart from factors such as patients' and physicians' expectations, physicians' training, and patient characteristics[7,8], the price and introduction of generics are potential drivers for overconsumption[9–14]. Generic drugs enter the market shortly after exclusivity rights (e.g., patent protection and data exclusivity) for the original patented drug end[14]. This was made possible by the Hatch-Waxman act, allowing companies to obtain approval for generic drugs without conducting additional clinical trials[14]. As a result, availability and access increase, while individual and national health expenditures fall, benefiting healthcare systems[15,16]. In the case of antibiotics, generics could potentially increase inappropriate antibiotic consumption and hasten the development of antibiotic resistance[9–14,17].

After generic entry, amoxicillin-clavulanate use in the United States and ciprofloxacin use in Denmark increased[9,14]. In Denmark, study findings showed a correlation between generic entry of ciprofloxacin and increased cases of ciprofloxacin-resistant *Escherichia coli* from urine isolates[9]. In Germany, generic entry of cephalosporins and fluoroquinolones increased the use of these antibiotic classes[13]. A more recent study examining the effect of generic entry of levofloxacin on fluoroquinolone use and meropenem on carbapenem use in five European Union countries and in the United States[18] showed significant increases in some cases and significant decreases in others, with no discernible patterns between different antibiotics or different countries. These previous studies have focused only on a few antibiotics, not addressed alternative explanations to their findings (co-interventions)[9,14,18] or not accounted for secular trends[14].

In this study, we analyze the effect of loss of exclusivity on the use of antibiotics for systemic use that had at least one generic enter the US market between 2000 and 2012, using an interrupted time series (ITS) design. We hypothesized that a level increase in antibiotic prescriptions would be visible by 6–12 months after the entry of the first generic and by 24 months at the latest. This was the case for five antibiotics (aztreonam, cefpodoxime, ciprofloxacin, levofloxacin, and ofloxacin), whereas one (cefdinir) showed a decrease, one year after generic entry. These changes were sustained two years after. Our second hypothesis was an increase in trend, which was the case for three antibiotics (cefprozil, cefuroxime axetil, and clarithromycin). These changes represented use leveling out. Together, these findings indicate no consistent pattern for antibiotic use following generic entry in the United States.

## Results

### Descriptive analysis
We identified 13 antibiotics that met the inclusion criteria (Figs. 1 and 2). Cefuroxime axetil was included despite generic availability of cefuroxime sodium, which comes only as an intravenous (IV) drug, whereas cefuroxime axetil is an oral treatment available for outpatient care. Azithromycin had the highest number of prescriptions—approximately three times greater than any other antibiotic—at the time of generic entry, followed by ciprofloxacin, cefdinir, levofloxacin, clarithromycin, and cefuroxime axetil. For all but two antibiotics (aztreonam and piperacillin/tazobactam), prescriptions almost completely consisted of generic products after generic entry. Aztreonam and piperacillin/tazobactam were, together with cefpodoxime, demeclocycline, and meropenem, the antibiotics with only one or two

manufacturers of generic products one year after generic entry (Supplementary Table 1 and Supplementary Fig. 1).

**Overview of results**. Prescriptions for five antibiotics—aztreonam, cefpodoxime, ciprofloxacin, levofloxacin, and ofloxacin—had a statistically significant increase 6–12 months after generic entry, which was sustained two years after generic entry. Cefdinir was the only antibiotic showing a significant decrease in prescriptions against the historical trend, sustained over time (Tables 1 and 2). Cefprozil, cefuroxime axetil, and clarithromycin experienced immediate significant increases in trend, but with no significant changes in the level detected within two years of generic entry. Eight of the 13 antibiotics had a declining trend prior to generic entry (Table 1) (9 including levofloxacin based on visual inspection), including four of the antibiotics with the highest number of prescriptions (cefuroxime axetil, ciprofloxacin, clarithromycin, and levofloxacin). In two of the cases showing significant level changes—ciprofloxacin and levofloxacin—prescriptions initially declined, then rose. In four cases—aztreonam, cefdinir, cefpodoxime, and ofloxacin—the changes represented trends leveling out. This was also the case for cefprozil, cefuroxime axetil, and clarithromycin. For the remaining antibiotics, no significant changes were detected (Figs. 1 and 2).

**Changes in number of prescriptions 6–12 months after generic entry**. Six months after generic entry, ciprofloxacin prescriptions had increased by 427.82 per one million population (95% confidence interval [CI] 247.18 to 608.45, $p$-value < 0.001, relative change 12%), ofloxacin prescriptions had increased significantly by 0.17 per one million population (95% CI 0.10 to 0.24, $p$-value < 0.001; note logged data, relative change 4%), whereas cefdinir prescriptions had decreased by −0.27 per one million population (95% CI −0.45 to −0.10, $p$-value 0.003; note logged data, relative change 3%). Twelve months after generic entry, aztreonam prescriptions had increased significantly by 0.04 per one million population (95% CI 0.01 to 0.08, $p$-value = 0.018, relative change 406%), cefpodoxime prescriptions increased by 0.22 per one million population (95% CI 0.03 to 0.41, $p$-value = 0.026; note logged data, relative change 5%), and levofloxacin prescriptions increased by 672.41 per one million population (95% CI 495.93 to 848.89, $p$-value < 0.001, relative chage 29%) (Table 2 and Supplementary Table 2).

**Changes in number of prescriptions 24 months after generic entry**. For the six antibiotics that showed significant increases/decreases 6–12 months after generic entry, changes were sustained over the subsequent year. Twenty-four months after generic entry, prescriptions had increased significantly for ciprofloxacin by 1104.26 per one million population (95% CI 887.76 to 1320.76, $p$-value < 0.001, relative change 33%), ofloxacin prescriptions by 0.38 per one million population (95% CI 0.29 to 0.48, $p$-value < 0.001, note logged data, relative change 12%), whereas cefdinir prescriptions had decreased significantly by −0.85 per one million population (95% CI −1.12 to −0.58, $p$-value < 0.001, note logged data, relative change 10%). Aztreonam prescriptions had significantly increased by 0.05 per one million population (95% CI 0.01 to 0.09, $p$-value = 0.008, relative change not availbale due to negative values), cefpodoxime prescriptions by 0.45 per million population (95% CI 0.20 to 0.70, $p$-value = 0.001; note logged data, relative change 12%), and levofloxacin prescriptions by 1989.14 per one million population (95% CI 1668.09 to 2310.19, $p$-value < 0.001, relative change 125%) (Table 2 and Supplementary Table 2).

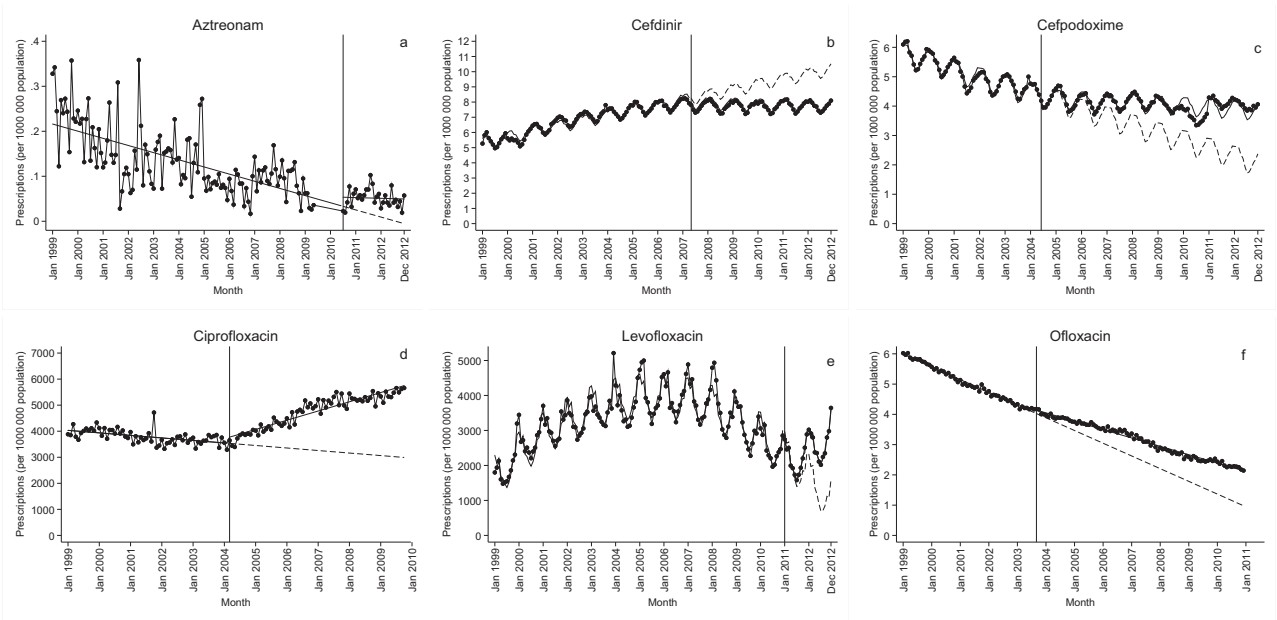

**Fig. 1 Antibiotics showing a significant level increase/decrease within two years after generic entry.** Change in number of antibiotic prescriptions per one million population before and after generic entry (vertical line), with projected level of prescriptions if generic entry had not taken place (dashed line). **a** Aztreonam. **b** Cefdinir; note logged data. **c** Cefpodoxime; note logged data. **d** Ciprofloxacin. **e** Levofloxacin. **f** Ofloxacin; note logged data.

**Fig. 2 Antibiotics showing no significant level change within two years after generic entry.** Change in number of antibiotic prescriptions per one million population before and after generic entry (vertical line), with projected level of prescriptions if generic entry had not taken place (dashed line). **a** Azithromycin. **b** Cefprozil; note logged data. **c** Cefuroxime axetil. **d** Clarithromycin. **e** Demeclocycline. **f** Meropenem. **g** Piperacillin/tazobactam.

**Table 1 Trends before and after generic entry.**

| Antibiotic | Baseline trend | | | Immediate change in trend | | |
|---|---|---|---|---|---|---|
| | Estimate | 95% CI | *p*-Value | Estimate | 95% CI | *p*-Value |
| Azithromycin | 37.991 | (22.219, 53.763) | <0.001 | −3.677 | (−25.545, 18.190) | 0.740 |
| Aztreonam | −0.001 | (−0.002, −0.001) | <0.001 | 0.001 | (−0.001, 0.004) | 0.328 |
| Cefdinir[a] | 0.028 | (0.023, 0.032) | <0.001 | −0.032 | (−0.042, −0.023) | <0.001 |
| Cefpodoxime[a] | −0.023 | (−0.029, −0.018) | <0.001 | 0.019 | (0.012, 0.027) | <0.001 |
| Cefprozil[a] | −0.014 | (−0.015, −0.013) | <0.001 | 0.008 | (0.006, 0.010) | <0.001 |
| Cefuroxime axetil | −18.691 | (−26.250, −11.132) | <0.001 | 17.513 | (9.466, 25.560) | <0.001 |
| Ciprofloxacin | −8.066 | (−11.648, −4.485) | <0.001 | 37.580 | (32.815, 42.346) | <0.001 |
| Clarithromycin | −28.233 | (−36.437, −20.028) | <0.001 | 17.987 | (6.922, 29.052) | 0.002 |
| Demeclocycline | 0.018 | (0.000, 0.035) | 0.044 | −0.012 | (−0.033, 0.009) | 0.251 |
| Levofloxacin | | | | 57.064 | (2.188, 111.940) | 0.042 |
| Meropenem | 0.001 | (0.000, 0.001) | 0.007 | 0.002 | (−0.002, 0.007) | 0.281 |
| Ofloxacin[a] | −0.035 | (−0.037, −0.033) | <0.001 | 0.012 | (0.010, 0.014) | <0.001 |
| Piperacillin/Tazobactam | −0.001 | (−0.002, 0.001) | 0.199 | 0.001 | (−0.008, 0.009) | 0.880 |

Results of the interrupted time series analysis, measuring the changes in number of antibiotic prescriptions per one million population per month. "Baseline trend" corresponds to the trend in prescriptions before generic entry. "Immediate change in trend" corresponds to the change in trend after generic entry. Graphs used to assess linearity and seasonality, as well as Akaike information criterion and the Bayesian information criterion for each antibiotic is available in the Supplementary Figs. 4–16. Either segmented regression was used with Prais–Winsten regression when autocorrelation was present. Two-sided test was used with no adjustment for multiple comparisons.
[a]Data were logged.

**Sensitivity analysis and co-interventions**. In 2001, the United States experienced the anthrax attacks. As a result, the United States stockpiled ciprofloxacin, leading to price negotiations with the innovator, Bayer[19,20]. After the US government threatened a compulsory license, Bayer agreed to lower prices in October 2001. Given the unique media attention and direct price intervention for ciprofloxacin, we conducted an ITS analysis excluding the anthrax attacks. This analysis showed that generic introduction had a significant increase in prescriptions of 408.60 per one million population (95% CI 59.54 to 757.66, *p*-value = 0.022, relative change 11%) after 12 months and 803.76 per one million population (95% CI 314.13 to 1293.39, *p*-value = 0.002, relative change 22%) by 24 months (Supplementary Tables 2–4). Meropenem and piperacillin/tazobactam showed greater variation in the beginning of the time series, whereas ciprofloxacin had one outlier in the pre-intervention data. In addition, we identified a total shortage of aztreonam between June 2009 and June 2010 reported by the Food and Drug Administration (FDA). Removing data for the months affected by the aztreonam shortage, removing the ciprofloxacin outlier, and restricting the dataset to only include data from 2005 in the case of meropenem and from 2007 in the case of piperacillin/tazobactam did not alter the results, apart from postponing the significant increase seen in aztreonam by 6 months (Supplementary Tables 3 and 4).

## Discussion

In this study, we used ITS analyses to measure the effect of generic entry on antibiotic prescriptions in the United States. Significant increases in prescriptions were observed one year after generics were introduced in the case of five antibiotics: aztreonam, cefpodoxime, ciprofloxacin, levofloxacin, and ofloxacin. One antibiotic—cefdinir—had a significant decrease in prescriptions one year after generic entry. For all six, the changes were sustained two years after. In the case of aztreonam, cefpodoxime, and ofloxacin, the changes represented negative trends leveling out. In two instances—ciprofloxacin and levofloxacin—generic entry led to negative prescription trends turning positive. Although cefprozil, cefuroxime axetil, and clarithromycin experienced immediate significant increases in trend, there were no significant changes in the level detected within two years of generic entry. Use of these three antibiotics represented a leveling

out over time. In the remaining cases, results were insignificant. All three fluoroquinolones included in the study were among the five antibiotics showing significant increases sustained two years after generic entry. However, the results suggest that the increase in ciprofloxacin was partly the result of the anthrax attacks and the subsequent price negotiation and stockpiling in October 2001.

Research on the effect of market exclusivity on drug use is relevant for the discussion about affordability and access to medicines, as well as responsible use. Multiple reports show how drug prices are increasing in the United States, reducing access and compromising health[21,22]. At the same time antibiotic resistance is increasing globally because of overuse of antibiotics[4]. For most antibiotics included in this study, use was declining even before generic entry took place and no dramatic change occurred when generics entered the market. This is not surprising, given that antibiotics are relatively inexpensive products, available only by prescription, and face competition from other antibiotics even when under patent protection. Many antibiotics also show strong seasonal trends, implying that the use depends on infectious disease prevalence—something that makes antibiotics different from, e.g., opioids.

Other factors coincident with generic entry, so-called co-interventions, could alter the trajectory of prescriptions[23,24]. We found that generic entry for aztreonam coincided with the market entry of Cayston (an inhalation treatment often used for patients with cystic fibrosis). This could have promoted the IV formulation of aztreonam as a treatment option for patients with cystic fibrosis and increased its use, as the IV formulation of aztreonam has clinical benefits for cystic fibrosis patients as well[25], and as IV antibiotics are sometimes used for preparing inhalations. Another plausible explanation for the observed changes in prescriptions for aztreonam, cefpodoxime, and ofloxacin is that consumption leveled out when approaching zero, as "negative consumption" is not possible. In the case of ciprofloxacin, the anthrax attacks and resulting stockpiling seem to have affected prescriptions. We did not find alternative explanations for the significant changes seen in the other antibiotics. FDA reported safety issues for fluoroquinolones in October and December 2008 due to an increased risk for tendinitis and tendon rupture, and azithromyzin in May 2012 and December 2013 due to an increased risk for fatal irregular heart rythms[26]. None of these events coincided with time of

**Table 2 Level change after generic entry.**

| Antibiotic | Change in level 6 months after generic introduction | | | Change in level 12 months after generic introduction | | | Change in level 18 months after generic introduction | | | Change in level 24 months after generic introduction | | |
|---|---|---|---|---|---|---|---|---|---|---|---|---|
| | Estimate | 95% CI | p-Value | Estimate | 95% CI | p-Value | Estimate | 95% CI | p-Value | Estimate | 95% CI | p-Value |
| Azithromycin | 182.165 | (−852.199, 1216.530) | 0.728 | 160.101 | (−902.728, 1222.929) | 0.766 | 138.036 | (−968.186, 1244.258) | 0.806 | 115.971 | (−1046.904, 1278.846) | 0.844 |
| Aztreonam | 0.034 | (−0.006, 0.074) | 0.100 | 0.042 | (0.007, 0.076) | 0.018 | 0.049 | (0.014, 0.084) | 0.006 | 0.052 | (0.010, 0.094) | 0.008 |
| Cefdinir[a] | −0.271 | (−0.447, −0.095) | 0.003 | −0.464 | (−0.659, −0.269) | <0.001 | −0.658 | (−0.885, −0.431) | <0.001 | −0.851 | (−1.118, −0.584) | <0.001 |
| Cefpodoxime[a] | 0.101 | (−0.067, 0.270) | 0.237 | 0.217 | (0.026, 0.407) | 0.026 | 0.333 | (0.113, 0.553) | 0.003 | 0.449 | (0.195, 0.702) | 0.001 |
| Cefprozil[a] | −0.076 | (−0.154, 0.002) | 0.057 | −0.028 | (−0.108, 0.052) | 0.491 | 0.020 | (−0.064, 0.103) | 0.644 | 0.067 | (−0.021, 0.156) | 0.134 |
| Cefuroxime axetil | −118.500 | (−306.791, 69.791) | 0.216 | −13.424 | (−236.923, 210.076) | 0.906 | 91.653 | (−171.241, 354.547) | 0.492 | 196.729 | (−108.126, 501.584) | 0.204 |
| Ciprofloxacin | 427.815 | (247.184, 608.446) | <0.001 | 653.298 | (464.246, 842.349) | <0.001 | 878.780 | (677.562, 1079.998) | <0.001 | 1104.263 | (887.763, 1320.762) | <0.001 |
| Clarithromycin | 162.386 | (−285.127, 609.899) | 0.475 | 270.309 | (−198.835, 739.453) | 0.257 | 378.232 | (−120.507, 876.970) | 0.136 | 486.154 | (−48.821, 1021.130) | 0.075 |
| Demeclocycline | −0.319 | (−1.034, 0.396) | 0.379 | −0.392 | (−1.170, 0.386) | 0.322 | −0.464 | (−1.319, 0.390) | 0.285 | −0.537 | (−1.479, 0.405) | 0.262 |
| Levofloxacin | 103.343 | (−77.097, 283.784) | 0.260 | 672.409 | (495.931, 848.886) | <0.001 | −237.630 | (−514.428, 39.168) | 0.092 | 1989.136 | (1668.087, 2310.185) | <0.001 |
| Meropenem | −0.012 | (−0.078, 0.055) | 0.729 | 0.002 | (−0.056, 0.061) | 0.936 | 0.016 | (−0.045, 0.077) | 0.598 | 0.030 | (−0.043, 0.104) | 0.414 |
| Ofloxacin[a] | 0.170 | (0.097, 0.243) | <0.001 | 0.241 | (0.162, 0.320) | <0.001 | 0.312 | (0.226, 0.399) | <0.001 | 0.383 | (0.288, 0.478) | <0.001 |
| Piperacillin/tazobactam | −0.045 | (−0.217, 0.127) | 0.606 | −0.041 | (−0.199, 0.116) | 0.606 | −0.037 | (−0.195, 0.120) | 0.641 | −0.034 | (−0.207, 0.140) | 0.702 |

Results of the interrupted time series analysis, measuring the change in number of antibiotic prescriptions per one million population 6, 12, 18, and 24 months after generic entry. Graphs used to assess linearity and seasonality, as well as Akaike information criterion and the Bayesian information criterion for each antibiotic is available in the Supplementary Figs. 4–16. Either segmented regression was used with Prais–Winsten regression when autocorrelation was present. Two-sided test was used with no adjustment for multiple comparison.
[a]Data were logged.

generic entry for these drugs. Unrelated to generic entry, it is possible that the decrease in levofloxacin, beginning in 2008, was partly due to safety issues, as that was the year FDA issued a safety warning. One of the effects of vaccines is a decrease in the use of antibiotics[27]. Five vaccines were added to the vaccination program in the United States between 2000 and 2012; pneumococcal vaccines in 2000 (7-valent) and 2010 (13-valent), influenza vaccine in 2004, quadrivalent meningococcal vaccine in 2005, and rotavirus vaccine in 2006[28]. The 7-valent pneumococcal vaccine had a major impact on infections caused by the bacteria strains covered by the vaccine, with the main effect seen between 2000 and 2002[29]. This coincided with the generic introduction of cefuroxime axetil and could have masked a significant increase in use, which, according to our results, had an insignificant increase. Based on the same logic, the increase in the use of levofloxacin could have been reduced by the introduction of the 13-valent pneumococcal vaccine causing a decrease in upper respiratory tract infections. However, decreases should then be visible in other respiratory tract antibiotics as well at this point in time, which was not the case. In contrast to the other vaccines, which target mainly the younger population, the influenza vaccine targets all age groups. Although this makes it relevant to this study, the impact of the influenza vaccine is more difficult to assess given that both effectiveness and coverage differs each season. Therefore, we cannot rule out that the effect of generic entry of an antibiotic used for respiratory tract infections was masked by the influenza vaccine. In 1995, the Centers for Disease Control and Prevention (CDC) launched the National Campaign for Appropriate Antibiotic Use in the Community, which was renamed Get Smart: Know when antibiotics work campaign in 2003[30,31]. Get Smart promotes responsible use of antibiotics and supports states and communities to develop and implement stewardship programs[32]. We did not find any programs that were implemented simultaneously in all states, and which could be linked in time with the dates for generic entry of the antibiotics included in this study. Although statewide and community-based programs have contributed to improvement in antibiotic use, it is unlikely that they would be able to cause a significant change in national use of one specific antibiotic at a specific point in time. Between 2000 and 2012, CDC reported increases of macrolide-resistant bacteria, including *Streptococcus pneumoniae*, Carbapenem-resistant Enterobacteriaceae, extended spectrum Betalactamase, and fluoroquinolone-resistant bacteria[2,33]. Although resistance develops gradually, an immediate impact on use could happen if the resistance levels led to nationwide guideline changes. We were not able to detect a guideline change that coincided with our dates for generic entry. On the contrary, all three fluoroquinolones included in the study were among the antibiotics with significant increases in prescriptions and in 2012 azithromycin was the most prescribed antibiotic in outpatient care in the United States, despite growing levels of resistance towards these antibiotics[34].

It is possible that the antibiotics included in this reseach impacted each other by competing for the same market shares. This would be relevant primarily for antibiotics with similar indications, formulation, and generic entry at close proximity in time. The three fluoroquinolones included in the study all showed an increase in use, so regardless of whether or not they had an impact on each other it should not have affected the outcome. However, a number of the oral antibiotics, where generic entry had a mixed effect, have respiratory tract infections as indications —azithromycin, cefdinir, cefpodoxime, cefprozil, cefuroxime axetil, clarithromycin, as well as ciprofloxacin, levofloxacin, and ofloxacin (Supplementary Table 5). It is possible that the generic entry of azithromycin, preceeding the generic entry of cefdinir and cefprozil, could have contributed to the lack of increase in cefprozil and the decrease of cefdinir. Then again, as

azithromycin use far exceeds the use of any of the other antibiotics, one would expect a similar effect on some of the other antibiotics, or that there would be no significant increases at all for any of the antibiotics with lower levels of use. Given the number of indications for each antibiotic, it is difficult to know exactly which antibiotic would substitute another in practice and we acknowledge that there is likely some level of impact between the different antibiotics. However, we were not able to detect any patterns that would support the idea that our findings were the result of generic entry of the other antibiotics.

The significant decrease in cefdinir was surprising. It may have been related to the decrease in the prevalence of acute otitis media, which has been attributed partly to a change in treatment recommendations and introduction of pneumococcal vaccines[35,36]. However, the decrease took place in between the introduction of the two vaccines, so it is unlikely that the 7-valent pneumococcal vaccine would have caused a sharp decrease in use seven years after introduction. Clinical treatment guidelines from 2004 lists amoxicillin as the first-line treatment for acute otitis media. In the case of allergy, cefdinir, cefpodoxime, and cefuroxime axetil are listed as alternatives[37]. It is possible that the generic introduction of cefuroxime axetil in 2002 and cefpodoxime in 2004 contributed to the decreased use of cefdinir. In addition, azithromycin, one of the most commonly used antibiotics for respiratory tract infections, became available as a generic little more than a year before cefdinir became available as a generic. This could have contributed to the decrease in cefdinir since the two antibiotics have similar indications.

Our findings suggest that, consistent with evidence from other countries[18], generic entry has limited and inconsistent effects on antibiotic use in the United States, with no significant, sustained increase in more than half of cases. However, we note that there was a positive significant change in trend without an accompanying change in level for three of the 13 antibiotics and, although trends leveling out as sales approach zero might partly explain this, we cannot rule out the possibility that introduction of generics may lead to a change in trend that leads to increased prescribing over longer time horizons. Overall, prescriptions were surprisingly stable over time. The reason for increasing levels of antibiotic prescriptions (and thereby use) and resistance is multifactorial and complex, and models should not assume that generic entry will automatically increase antibiotic prescriptions in settings like the United States. Nevertheless, some antibiotic classes, namely fluoroquinolones, could be more sensitive. Interestingly, a significant increase in fluoroquinolone use was also observed by Stephens[18], by Jensen et al.[9], and by Kaier[13]. However, it is difficult to know whether this is coincidental, as the three studies did not include more than one other antibiotic or class and did not discuss alternative explanations for their findings. Also, the increase in fluoroquinolones was notable, given the well-known problems related to both resistance and side effects[38]. Arguably, this increase could be considered inappropriate (even if the anthrax attacks were a likely contributing factor), but without looking at overall antibiotic use and use by antibiotic class, this is difficult to judge.

Our study included all antibiotics for which generics were introduced between 2000 and 2012, which allows for analyzing differences among classes, formulations, and target indications. Nevertheless, several limitations should be considered when interpreting our findings. Research shows that price reductions depend on the number of generics introduced to market[17], with a small effect on price after the first generic entry[39]; the biggest price reduction occurs after the second generic entry[40]. We did not have access to dates of subsequent generic entry. However, for eight of the 13 antibiotics in our study, three or more generic products were approved one year after the first generic entry (Supplementary Table 1)[38], making multiple generic entries a

possibility for most of the antibiotics. In addition, for most antibiotics, generics accounted for the majority of prescriptions within one year of their introduction (Supplementary Fig. 1). According to the Hatch-Waxman Act, the first generic product is allowed 180 days of exclusivity against subsequent generic entrants[14]. Although the rule does not cover authorized generic products (generics produced by the brand owner), it could have delayed a potential price reduction by limiting competition. To account for a delay in the effect of generic entry, we included analysis of the effect 24 months after the event. Information regarding the number of manufacturers includes only approvals to market and depends on companies actively notifying the FDA if products are taken off market. This means that the number of manufacturers could have been overestimated. We did not examine class-based substitution effects, which could explain some of the trends observed—e.g., if increases in the use of a particular antibiotic were due to declines in sales in a substitutable product. Finally, we addressed the issue of co-interventions by searching for events that could have affected use. However, given the multitude of factors that influence antibiotic use, there could be co-interventions that we have overlooked. Additional research is needed to explore the effect of generic entry on antibiotic prices and to consider differences among antibiotic classes. Research is also needed to determine the effects, including access and antibiotic resistance, when generics enter the market in low- and middle-income countries.

## Methods

**Study design**. We used an ITS design, a quasi-experimental approach to evaluating public health interventions[23,24], to assess the effect of loss of market exclusivity on antibiotic use. The intervention being studied was generic entry, defined as the month when prescriptions of the first generic were first recorded. We chose generic entry to mark the end of exclusivity because alternatives, such as patent expiration date or approval date of the first generic product, do not guarantee that a generic has entered the market. Moreover, companies manufacturing brand-name antibiotics may contract with generic manufacturers to distribute a generic version of the product prior to the end of exclusivity. We therefore compared the date of the first recorded generic prescription with FDA approval dates to ensure that approval preceded distribution of generics. This was the case for all but one antibiotic (ciprofloxacin), for which the intervention date was set to the month when both generic prescription and approval had occurred (Supplementary table 1). The outcome measure was total number of prescriptions per capita per month (including oral and parenteral formulations of both brand and generic products).

The effect estimates used in ITS analysis is level and trend[23]. With respect to antibiotics, a level change represents the difference in antibiotic use between the specified post-intervention time point and the pre-intervention regression line that is extrapolated to that same time point (counterfactual). A change in trend represent an increase or decrease in the slope of a time series segment after the intervention compared with the time series segment preceding the intervention[23] (Supplementary Fig. 2). There is no consensus on which effect estimate to report, and they are inconsistently used when reporting the results of ITS analysis[41]. The selection of which effect estimates to give weight ultimately depend on the specifics of each individual study, including the intervention and its expected impact on the outcome measure. Research show that the number of generic products entering the market and generic price decline is biggest during the first 12–24 months after generic entry, and stabilizing after that[42]. Therefore, it is reasonable to assume that an impact on antibiotic prescriptions should be visible by 6–12 months after the entry of the first generic. Thus, our pre-specified hypothesis was a change in level at 6 and 12 months as the primary time points to model the effect. To assess whether the effect was consistent over a longer period or delayed, we also modeled the effects after 18 and 24 months. These are the same or similar time points that have been used in other research measuring the impact of generic entry on antibiotic use[9,14,18]. In addition, we examined the trend after first generic entry to assess changes in the rate of antibiotic prescriptions. Our second hypothesis was an immediate change in trend.

**Data sources**. Data on antibiotic prescriptions in the United States were obtained from the IQVIA Xponent database, which contains the monthly number of antibiotic prescriptions filled in a pharmacy based on product sales data from retail pharmacies. IQVIA Xponent data have been extensively used in other studies of antibiotic prescribing[43–47]. As all prescriptions that are filled are not necessarily consumed, the number of filled prescriptions may overstate consumption. We accounted for population growth using data from the US Census Bureau[48].

**Identifying antibiotics for analysis**. We identified antibiotics of interest by reviewing monthly product prescription data. The inclusion criteria were antibiotics for systemic use for which a generic product entered the market between 2000 and 2012 of the same formulation and strength based on New Drug Application (NDA)/Abbreviated New Drug Application (ANDA) status, obtained from the FDA website[38]. The following antibiotics were included: azithromycin, aztreonam, cefdinir, cefpodoxime, cefprozil, cefuroxime axetil, ciprofloxacin, clarithromycin, demeclocycline, levofloxacin, meropenem, ofloxacin, and piperacillin/tazobactam (Figs. 1 and 2). All antibiotics included in our analysis had sales data for at least 24 months before and 24 months after the date of generic entry, timelines started in January 1999 for all antibiotics. Additional information about each antibiotic, obtained from the FDA website[38], included the following: active substance, brand name, generic name, company responsible for approval of the brand drug, date of NDA/ANDA approval, formulation, class, approved indication, and number of companies with approved generics 12 and 24 months after generic entry (Supplementary Tables 1 and 5).

**Statistical analysis**. Segmented regression analysis, a statistical method commonly used in ITS design, was conducted and uses time-series data to measure change in level and change in trend, allowing for underlying trends before and after the event of interest (intervention)[23,24,49]. The statistical model used for this model is

$$Y_t = \beta_0 + \beta_1 * \text{Time}_t + \beta_2 * \text{Intervention}_t + \beta_3 * \text{Time after intervention}_t + e_t \quad (1)$$

where $Y_t$ is the total number of prescriptions per capita per month at time $t$, $\beta_0$ estimates the baseline level, $\beta_1$ estimates the pre-intervention trend, $\beta_2$ estimates the change in level, and $\beta_3$ is the change in trend. "Time" is a continuous variable and is time since the start of the series. "Intervention" is an indicator for when the intervention occurred at time $t$ and is coded 0 pre-intervention and 1 post-intervention. "Time after intervention" is a continuous variable, time $t$ is coded 0 pre-intervention and is time − time point when the intervention occurred. $e_t$ is the error term. Supplementary Fig. 2 shows a description of the effect estimates. Autocorrelation and seasonality were assessed using autocorrelation function and partial autocorrelation function. Visual inspection, as well as looking at the predicted values, were also used to determine whether seasonality needed to be adjusted for, as well as whether to fit a linear or a quadratic regression, or if the data needed to be log-transformed (Supplementary Table 6). If autocorrelation was present, a Prais–Winsten regression was used, which assumes first-order autocorrelation process. If seasonality was present, then a seasonal covariate was included. To assess the model fits, the Akaike information criterion and the Bayesian information criterion were used. The level of significance was set to 5%. Outliers were addressed through sensitivity analysis by assessing whether results were robust to their exclusion. Time series were complete for all but one antibiotic (aztreonam), which lacked two months of data. This is possibly because of failure in the reporting system or low sales. Because of deviant data points (not related to the generic entry, as they occurred more than six years after), the timelines for ciprofloxacin and ofloxacin were shortened (Supplementary Fig. 3). In one case (levofloxacin), quadratic regression was fitted, resulting in no baseline estimation (Supplementary Table 6). In addition to absolute change, relative change was calculated using the following equation:

$$R_t = 100 * \frac{\text{Level effect}_t}{\left(\text{Predicted value}_t\right) - \left(\text{Level effect}_t\right)} \quad (2)$$

where $R_t$ is the relative change at time $t$, "Level effect$_t$" is the change in level at time $t$, and "Predicted value$_t$" is the counterfactual at time $t$[50]. Analysis was done using Stata15[51]. To identify potential co-interventions (events that coincide with the intervention and affect the outcome measure thereby impacting the result), we conducted a systematic search using open-access internet resources and published articles[23,24]. Factors considered possible co-interventions included safety issues, infectious disease outbreaks, changes in treatment guidelines, vaccination programs, stockpiling, drug shortages, price negotiations, stewardship programs, and antibiotic resistance. Safety issues for each of the 13 antibiotics were identified using the FDA drug safety communications[26]. Infectious disease outbreaks and development of antibiotic resistance were identified searching reports by the CDC[2,33]. We searched for changes in treatment guidelines developed by the Infectious Diseases Society of America and in treatment guidelines posted on the CDC webpage[52,53]. Changes in the US vaccination program were identified using the CDC "Immunization schedules"[28]. In addition, we searched PubMed Google and Google Scholar using the following combinations of search terms: [pathogen] AND vaccine; outbreak bacteria USA; antibiotic prescriptions USA; antibiotic stewardship program AND [antibiotic]; antibiotic shortage; antibacterial drug shortages; antibiotic stockpiling; antibacterial stockpiling; price negotiations AND [antibiotic].

**Reporting summary**. Further information on research design is available in the Nature Research Reporting Summary linked to this article.

## Data availability

The data that support the findings of this study are available from IQVIA (www.iqvia.com), but restrictions apply to the availability of these data, which were used under license for the current study, and so are not publically available. The terms of IQVIA's licensing agreement preclude sharing of the data. Other researchers may purchase the data from IQVIA directly.

## Code availability

STATA code is available from Github under https://github.com/ceka2000/Antibiotic-codes/tree/v1.0.0

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

## Acknowledgements

C.K. and C.Å. were funded by the DRIVE-AB Consortium. DRIVE-AB is supported by the IMI Joint Undertaking under the DRIVE-AB grant agreement number 115618, the resources of which are composed of financial contributions from the European Union's Seventh Framework Programme and the European Federation of Pharmaceutical Industries and Associations companies' in-kind contribution. C.K. and C.Å. were partly supported by the Research Council of Norway through the Global Health and Vacci-nation Programme (GLOBVAC), project number 234608. K.O. is supported by NOA 06-IDSET160030 from the Biomedical Advanced Research and Development Authority (BARDA) under the Assistant Secretary for Preparedness and Response (ASPR) in the US Department of Health and Human Services, and the CARB-X award from the Wellcome Trust, but the views expressed herein are not necessarily those of CARB-X or any CARB-X funder. J.H. works for Health Services Research Unit, University of Aberdeen, and is core funded by the Chief Scientist Office of the Scottish Government Health and Social Care Directorates. R.L. was supported by 16IPA16092427 from the US Centers for Disease Control and Prevention. The funder provided support in the form of salaries for authors, according to the statement above, but did not have any additional role in the study design, data collection and analysis, decision to publish, or preparation of the manuscript.

## Author contributions

C.K., C.Å., and J.-A.R. designed the study. R.L. and E.K. provided the data. J.H. and C.K. conducted the analysis and drafted the article. H.S.B., C.Å., E.K., M.L., K.O., J.-A.R., and R.L. reviewed and contributed to multiple revisions and approved the final version of the article.

## Competing interests

The authors declare no competing interest.
