## [Peer Review File · Nature Communications]

REVIEWER COMMENTS

Reviewer #1 (Remarks to the Author):

The authors report an interrupted time series analysis, using monthly prescription data derived from the proprietary IQVIA dataset for the United States. The exposure of interest was the date of generic entry for each antimicrobial of interest. The outcome was monthly prescriptions per capita of that antimicrobial post-generic entry [primary timeline to 6-12 months post-generic entry]. The introduction of generic drugs on the market has benefits related to decreasing health care system costs and improving drug access, but the authors hypothesized that transition to generics may lead to increases in antimicrobial use (which could thereby increase antimicrobial resistance pressure).

Comments/Questions:

-the authors identified 13 antibiotics that met the inclusion criteria, but the main display (Figure 1) only includes antibiotics with a 'significant change in number of antibiotic prescriptions'. Wouldn't it be more appropriate for Figure 1 to include all 13 agents? Why are the other 6 agents relegated to the supplemental material?

-would a combined analysis across the 13 drugs be worthwhile?

-would it be interesting to look at negative tracer drugs from the same classes as these 13 agents that did not undergo generic drug entry during the study period? For example, another fluoroquinolone, another cephalosporin etc.? Would it be worth looking at some that were exclusive during the entire period? Would it be worth looking at some that were generic during the entire period?

-Line 131-134"finally we conducted a search using open-access internet resources and published articles, including the FDA website, US Centers for Disease Control, ...to identify potential co-interventions such as outbreaks, indications, stockpiling, shortages, priced negotiations and safety warnings"...

-...was this planned a priori, or was this conducted post-hoc in attempt to explain the widely varying trends for different agents.

-...was this search systematic?

Reviewer #2 (Remarks to the Author):

In this study, the authors explore the relationship between generic market entry for 13 antibiotics from 2000 to 2012. The research question is an interesting one from an economic perspective and the authors do an excellent job of presenting it. However, I felt the larger public health implications were unclear as many factors in addition to cost influence antibiotic prescribing patterns.

My major concern with this study is that it does not adequately address the role of other secular trends in antibiotic prescribing patterns. The authors briefly mention PCV7 and PCV13 as explanations for the decrease in cefdinir prescriptions, but I would urge them to consider the role of universal influenza vaccination and PCV on all drugs commonly used for respiratory infections (including cefdinir, cefpodoxime, and levo). Additionally, the authors should consider antibiotic resistance trends and if those could have had any effect on prescribing patterns during the 12+ year study period. For example, there was a marked increase in cipro that the authors attributed to anthrax stockpiling, but it is also worth considering if resistance patterns to other drugs commonly used for STIs and UTIs changed during that period and could have contributed to this increase. Finally, the authors mention stewardship efforts in the introduction, but do not fully consider the implications of a growing stewardship movement, especially in pediatrics during the study timeframe in the discussion. Pediatric stewardship could have accounted for some of the decreases in drugs commonly used in children (e.g. cefdinir and cefpodoxime).

Additional comments:

- Overall, the manuscript is well-written and clear, with a few exceptions noted below.
- It was unclear to me which drugs were included until I looked at the tables. It would be helpful to provide a list of the included drugs either in the introduction and/or methods section.
- It could be helpful to the reader to briefly include information on the IQVIA Xponent sample and

projection methodology.

- I had trouble understanding some elements of the methods; it would be helpful to clarify:
 - o Did you include all antibiotic formulations (oral, parenteral, otic, topical) or only systemic antibiotics?
 - o What was the timeframe included for the pre-generic introduction phase? Based on the figures it looks like the timeframes for all drugs started in 1999, but the methods do not make this clear and it seems like a standardized pre- period would make more sense.
- I am not familiar with the statistical methods used and will defer to additional statistical review.
- The authors do a nice job of considering the implications of antibiotic use on resistance in the introduction and discussion, but it might also be helpful to consider antibiotic-associated adverse events. For example, the sharp increase in cipro is notable as fluoroquinolones have been associated with severe adverse events.
- Introduction, line 58, citation #6: I would recommend citing the original study rather than the Pew report. I did not check if this was an issue for other references, but, when possible, please cite the original studies.
- Please check all references, it seems like the formatting may be off for some.
- Very minor comment – I think there is a typo in the abstract background section (lines 32-33).

Reviewer #3 (Remarks to the Author):

The effect of generic entry on antibiotic prescriptions in the United States, 2000-2012: An interrupted time series study

Summary of Paper

Antibiotic resistance is a major threat to public health, driven in part by overconsumption of antibiotics. When patents expire, branded antibiotics lose market exclusivity and generics typically enter the market at lower prices. This could potentially increase consumption of these antibiotics. This paper aims to examine the effect of generic market entry on antibiotic consumption in the United States. The method adopted is to use IQVIA data to conduct an interrupted time series analysis, spanning 2000-2012, of the change in volume of prescriptions per month per million people for antibiotics for which at least one generic entered the market. 13 different antibiotics, from several classes, are analysed. The paper reports that, one year after generic entry, prescriptions increased for 5/13 antibiotics, decreased for 1/13 and that there was no significant change for 7/13 antibiotics. The authors conclude that, in the United States, there is no consistent pattern for antibiotic use following generic entry.

This study addresses an important issue. Antibiotic resistance is a major issue in the US and beyond, but so too is the rising cost of health care and related issues of access to medicines. By lowering prices, the entry of generic antibiotics could potentially reduce health care costs and improve access, but at the possible cost of fuelling antibiotic resistance by increasing overall antibiotic consumption. As evidence on this topic is limited, the paper could potentially make a valuable contribution. For example, if strong evidence could be found that generic entry is unlikely to increase antibiotic consumption, this would be useful to know as it would suggest that the main result from generic entry is positive (lower health care costs). On the other hand, if we were to learn that generic entry appears to increase antibiotic consumption, this could spur important future research on whether the increased antibiotic consumption following generic entry is due to meeting previously unmet needs or to overconsumption.

However, I have a number of concerns about the paper as it currently stands, some of which are major.

Major points to consider:

1. In my view both the Statistical analysis subsection on pages 5-6, and the results themselves, need to be clearer and much more detailed (even if some details are deferred to supplementary materials). Some specific points:

a) A brief description should be available of what segmented regression and Prais-Winsten regression are.

b) We are told in lines 119-121 (p. 5) that "...the fitted values from the two models were compared graphically and by looking at the Akaike information criterion and the Bayesian information criterion."

I am not clear which two models are being referred to? In any case, it would be good to be able to see these graphs and the AIC/BIC (i.e. in the Results or, probably better, the supplementary materials).

c) In lines 121-123 we are told that "Visual inspection, as well as looking at the predicted values, determined whether seasonality was adjusted for, as well as whether to fit a linear or a quadratic regression..."

Again, it would be good to be able to see the graphs on which these decisions were made. If seasonality was not adequately adjusted for in initial models, then it would be good to read how was this addressed in the final models.

d) Similarly in lines 123-124, we are told that "outliers were addressed through sensitivity analysis by assessing whether results were robust to their exclusion." I think these sensitivity analyses should be available to the reader in either the Results or supplementary materials.

e) Overall, it is not sufficiently clear, with sufficient detail, what the specifications of the various models used were or how they were chosen.

2. I have a major concern with the interpretation of the results. The discussion of the results in the paper focuses almost entirely around the changes in the prescription levels in the 13 antibiotics after generic entry (as outlined in Table 2). The paper reports for example that one year after generic entry, prescriptions increased for 5/13 antibiotics, decreased for 1/13 and that there was no significant change for 7/13 antibiotics. The authors conclude from this that there is no consistent pattern for antibiotic use following generic entry. However, it seems to me that just as important as the changes in levels following generic entry are the changes in the trend, which was significantly positive for 7 of the 13 antibiotics (Table 1). These include Cefprozil, Cefuroxime axetil and Clarithromycin, all of which are included in Figure 3 and branded "antibiotics with no significant change". Looking at the more recent prescription volumes in Figure 3 for these 3 antibiotics, and comparing them to the extrapolated pre generic trends, it does appear that generic entry likely led to greater consumption than there would otherwise have been. From the perspective of considering an 'increase' following generic entry as meaning 'an increase in level with no decrease in the trend OR an increase in trend with no decrease in level' and taking a timepoint of 12 months following entry to assess the level change, 8/13 antibiotics exhibit an increase (Axtreonam, Cefpodoxime, Ciprofloxacin, Levofloxacin, Ofloxacin, Cefprozil, Cefuroxime axetil and Clarithromycin).

Now consider these results in the context of the issue you raised on p. 13 regarding research suggesting that "the biggest price reduction occurs after the second generic entry" (lines 277-278), and (lines 279-280) that "...for eight of the 13 antibiotics in our study, three or more generic products were approved one year after first generic entry (Appendix Table S1". Now looking at Table S1 and the 8 antibiotics that had 3 or more generics 1 year after generic entry, 6 of these antibiotics (Cefprozil, Cefuroxime axetil, Ciprofloxacin, Clarithromycin, Levofloxacin and Ofloxacin) increased following generic entry according to the criteria I suggest above.

With all this in mind, wouldn't it be fairer to conclude that your results provide at least tentative evidence to suggest that there tends to be an overall positive effect of generic entry on antibiotic consumption in the US?

3. I have some queries and comments on the calculations on absolute and relative changes in prescriptions (which I agree are very valuable to look at, including at the later timepoint of 24

months).

Firstly, I am a bit confused on how the calculations have been performed in places. For example, in the case of Levofloxacin, the change in level at 12 months was 672.41/m. We are told in line 178 (p. 8) that this is a relative change of 29%. This suggests that the base level was 2318.66/m. But in that case, surely the 1989.14/m increase at 24 months is an 86% of increase on this base, not a 125% increase as stated in line 190?

More broadly, in line with my concerns in point 2, when analysing the changes after 12 and 24 months, I think you should also consider the total changes, based on both the change in the level and the change in the trend (i.e. compare the size of the increases against the projected values at these time points based on the pre generic entry trends).

Minor points to consider:

1. Please consider adding a newer reference to supplement references 3-4, i.e. Chatterjee A, Modarai M, Naylor NR, et al. Quantifying drivers of antibiotic resistance in humans: a systematic review. *Lancet Infect Dis.* 2018;18:e368–78.
2. Please explain/define all acronyms before using (E.g. on p.5, NDA/ANDA; on p. 6, IV)
3. On page 6, lines 142-143 under the Descriptive analysis subsection, we read "Nine of the 13 antibiotics had a declining trend prior to generic entry (Table 1)),..."
Maybe this is a bit pedantic, but the trends in the table are from a model (except possibly for Levofloxacin) so perhaps this shouldn't go in this subsection. As an aside, is the declining trend for Levofloxacin from inspection of the inverted U-shape in Fig 1 or from a model?
4. On p. 8, line 181 – I think you mean 'five' rather than 'six' antibiotics here.
5. Identifying all other events (e.g. co-interventions) that may have influenced the results is of course infeasible/impossible. It is good that the study made some attempt to consider at least some major possible confounders. I would only suggest that, for full transparency and to give the reader more comfort, it would be good to provide a bit more detail on the methods you used, such as the search terms in PubMed, Google Scholar etc.
6. Perhaps include the sensitivity analysis described on p. 9 lines 195-197 in supplementary materials.
7. It is perhaps worth being clearer on explaining which direction the authors feel the market entry of Caystonem is more likely to have affected volume of Aztreonam (or stating that either direction seems plausible).
8. In the Discussion, on p. 11 line 230 – perhaps I would weaken the statement that "...the increase in ciprofloxacin was partly the result of the anthrax scare..." to something like "...the results suggest that the increase in ciprofloxacin was partly the result of the anthrax scare..." as I don't think we can be sure about this.
9. On p. 11 line 237, why is the fact that an antibiotic's use was declining before generic entry a sign that lack of access is not an issue?
10. Related to major point 2: on p. 12 line 259, it feels misleading to highlight that there only two cases of an upward trend after generic entry – at least without also mentioning that the change in the trend is positive in 7 cases.
11. On p. 13, lines 273-275 it is claimed that the study's approach "rules out the possibility that simultaneous generic entry of other antibiotics affected the antibiotics under consideration." Surely this isn't really true as each antibiotic was modelled separately in the study?

Signed: Laurence Roope

REVIEWER COMMENTS

Reviewer #1 (Remarks to the Author):

Reviewer comment: -the authors identified 13 antibiotics that met the inclusion criteria, but the main display (Figure 1) only includes antibiotics with a 'significant change in number of antibiotic prescriptions'. Wouldn't it be more appropriate for Figure 1 to include all 13 agents? Why are the other 6 agents relegated to the supplemental material?

Response: We have now added all 13 antibiotics included in the study in Figure 1.

Reviewer comment: -Line 131-134"finally we conducted a search using open-access internet resources and published articles, including the FDA website, US Centers for Disease Control, ...to identify potential co-interventions such as outbreaks, indications, stockpiling, shortages, priced negotiations and safety warnings"...

-...was this planned a priori, or was this conducted post-hoc in attempt to explain the widely varying trends for different agents.

-...was this search systematic?

Response: We agree that this needs to be clarified and have made additions to the methods section to elaborate on what was done to address co-interventions. We have also examined additional co-interventions, as suggested by the other reviewers, which has been added to the results section and discussion. Finally, we have clarified that the search was systematic and conducted after we finished the ITS analyses, see pages 7, lines 166-182 (method and material), pages 12-13, lines 257-308 (results), pages 16-17, lines 347-372 (discussion).

Reviewer comment: -would a combined analysis across the 13 drugs be worthwhile?

Response: Thank you for your comment. The effect estimate of a combined analysis across the 13 drugs would be hard to interpret since an increase in one antibiotic could be masked by a decrease in another. It would therefore be difficult to know if and to what extent each antibiotic contributed to the result. This in turn would make it difficult to analyse the results, including discussing the impact of co-interventions. We believe that individual analysis would allow for a deeper understanding of generic entry.

Reviewer comment: -would it be interesting to look at negative tracer drugs from the same classes as these 13 agents that did not undergo generic drug entry during the study period? For example, another fluoroquinolone, another cephalosporin etc.? Would it be worth looking at some that were exclusive during the entire period? Would it be worth looking at some that were generic during the entire period?

Response: Thank you for your comment and suggestion. We agree that using a comparator in a controlled ITS design can have value strengthening inference about the effect of generic entry. The challenge is to identify antibiotics that can serve as a reasonable comparator. Antibiotics with vastly different clinical indications does not serve as a valuable comparator, as they will be sensitive to different co-interventions, and likely have different underlying trends. Another option is to select antibiotics with similar clinical indications but these compete for the same market. Changes in one antibiotic will likely have an opposite effect on another of a similar class, and would be expected. The issue is further complicated by the fact that antibiotics often have multiple indications, and use to some extent depend on local

traditions, leading to uncertainties of whether or not the chosen comparator actually is a good comparator even if it belongs to the same class. This makes it difficult to draw conclusions from an analysis of a comparator and ultimately leads to speculation, which is why we have chosen to conduct our analysis without a comparator.

Reviewer #2 (Remarks to the Author):

Reviewer comment: My major concern with this study is that it does not adequately address the role of other secular trends in antibiotic prescribing patterns. The authors briefly mention PCV7 and PCV13 as explanations for the decrease in cefdinir prescriptions, but I would urge them to consider the role of universal influenza vaccination and PCV on all drugs commonly used for respiratory infections (including cefdinir, cefpodoxime, and levo).

Additionally, the authors should consider antibiotic resistance trends and if those could have had any effect on prescribing patterns during the 12+ year study period. For example, there was a marked increase in cipro that the authors attributed to anthrax stockpiling, but it is also worth considering if resistance patterns to other drugs commonly used for STIs and UTIs changed during that period and could have contributed to this increase.

Finally, the authors mention stewardship efforts in the introduction, but do not fully consider the implications of a growing stewardship movement, especially in pediatrics during the study timeframe in the discussion. Pediatric stewardship could have accounted for some of the decreases in drugs commonly used in children (e.g. cefdinir and cefpodoxime).

Response: The important issue of co-interventions has been raised by other reviewers as well, and we have now made additions to the methods section to elaborate on what was done to address co-interventions. We have also looked at some additional co-interventions, including a more extensive search on vaccines, antibiotic resistance, and stewardship interventions. The results of this search has been added to the results section and discussion. However, stewardship and resistance levels are not events that affects an entire nation at a specific point in time, but rather has a gradual impact since stewardship programs are often local/regional and takes time to implement. Because of this their impact is to some extent accounted for in the model by the underlying trend. Nevertheless, if the event (resistance level or stewardship) caused an instantaneous event that coincided with generic entry it could have impacted the results. We have attempted to identify events such as these, but in the manuscript we discuss that this is a difficult task to accomplish given that both stewardship programs, treatment guidelines and resistance levels can differ between regions and states, and since multiple antibiotics from different classes have the same indications. See pages 7, lines 166-182 (method and material), pages 12-13, lines 257-308 (results), pages 16-17, lines 347-372 (discussion).

Additional comments:

Reviewer comment: It was unclear to me which drugs were included until I looked at the tables. It would be helpful to provide a list of the included drugs either in the introduction and/or methods section.

Response: A list of antibiotics included in the study have now been added to the method section, page 5 lines 119-122.

Reviewer comment: It could be helpful to the reader to briefly include information on the IQVIA Xponent sample and projection methodology.

Response: We have made an addition in the methods section describing the IQVIA Xponent methodology, page 5 lines 107-110.

Reviewer comment: I had trouble understanding some elements of the methods; it would be helpful to clarify:

o Did you include all antibiotic formulations (oral, parenteral, otic, topical) or only systemic antibiotics?

Response: We have made an addition to clarify this, see page 4, line 99.

Reviewer comment: What was the timeframe included for the pre-generic introduction phase? Based on the figures it looks like the timeframes for all drugs started in 1999, but the methods do not make this clear and it seems like a standardized pre- period would make more sense.

Response: The timeframe was set to 1999 for all antibiotics. This allowed us to get as much information as possible about the pre-intervention trend, as well as conduct sensitivity analysis when there seemed to be outliers in the data. We have made additions to clarify that all timelines start from January 1999, page 5, lines 123-124.

Reviewer comment: The authors do a nice job of considering the implications of antibiotic use on resistance in the introduction and discussion, but it might also be helpful to consider antibiotic-associated adverse events. For example, the sharp increase in cipro is notable as fluoroquinolones have been associated with severe adverse events.

Response: Antibiotic-associated adverse events falls under co-interventions. As stated above, we have clarified and elaborated on this topic, including searching for and discussing the impact of safety issues, see pages 7, lines 166-182 (method and material), pages 12-13, lines 257-308 (results), pages 16-17, lines 347-372 (discussion).

Reviewer comment: Introduction, line 58, citation #6: I would recommend citing the original study rather than the Pew report. I did not check if this was an issue for other references, but, when possible, please cite the original studies.

Response: Thank you for bringing this to our attention. We have updated the reference, see page 3 line 60.

Reviewer comment: Please check all references, it seems like the formatting may be off for some.

Response: References have now been updated.

Reviewer comment: Very minor comment – I think there is a typo in the abstract background section (lines 32-33).

Response: We have removed “and” from line 32 on page 2.

Reviewer #3 (Remarks to the Author):

Reviewer comment: Major points to consider: 1. In my view both the Statistical analysis subsection

on pages 5-6, and the results themselves, need to be clearer and much more detailed (even if some details are deferred to supplementary materials). Some specific points:

a) A brief description should be available of what segmented regression and Prais-Winsten regression are.

Response: Thank you for your comment. We have rewritten the section named “Statistical analysis” in the method section to further describe and clarify statistical methods used, see page 6, lines 130-148. In addition, lines 86-89 page 4 have been integrated in the rewritten section.

Reviewer comment: b) We are told in lines 119-121 (p. 5) that “...the fitted values from the two models were compared graphically and by looking at the Akaike information criterion and the Bayesian information criterion.”

I am not clear which two models are being referred to? In any case, it would be good to be able to see these graphs and the AIC/BIC (i.e. in the Results or, probably better, the supplementary materials).

Response: We have added this material in the supplementary materials, figures S4-S16.

Reviewer comment: c) In lines 121-123 we are told that “Visual inspection, as well as looking at the predicted values, determined whether seasonality was adjusted for, as well as whether to fit a linear or a quadratic regression...”

Again, it would be good to be able to see the graphs on which these decisions were made. If seasonality was not adequately adjusted for in initial models, then it would be good to read how was this addressed in the final models.

Response: We have added this material in the supplementary materials, figures S4-S16. Information about how we addressed seasonality has been added on page 6 lines 145-148.

Reviewer comment: d) Similarly in lines 123-124, we are told that “outliers were addressed through sensitivity analysis by assessing whether results were robust to their exclusion.” I think these sensitivity analyses should be available to the reader in either the Results or supplementary materials.

Response: We have now added the results from the sensitivity analysis in the supplementary material, Table S5 and S6.

Reviewer comment: e) Overall, it is not sufficiently clear, with sufficient detail, what the specifications of the various models used were or how they were chosen.

Response: We have rewritten the section named “Statistical analysis” to describe the methods in more detail, see page 6 lines 130-148. Information on final models can be found in Figure S3.

Reviewer comment: 2. I have a major concern with the interpretation of the results. The discussion of the results in the paper focuses almost entirely around the changes in the prescription levels in the 13 antibiotics after generic entry (as outlined in Table 2). The paper reports for example that one year after generic entry, prescriptions increased for 5/13 antibiotics, decreased for 1/13 and that there was no significant change for 7/13 antibiotics. The authors conclude from this that there is no consistent pattern for antibiotic use following generic entry. However, it seems to me that just as important as the

changes in levels following generic entry are the changes in the trend, which was significantly positive for 7 of the 13 antibiotics (Table 1). These include Cefprozil, Cefuroxime axetil and Clarithromycin, all of which are included in Figure 3 and branded “antibiotics with no significant change”. Looking at the more recent prescription volumes in Figure 3 for these 3 antibiotics, and comparing them to the extrapolated pre generic trends, it does appear that generic entry likely led to greater consumption than there would otherwise have been. From the perspective of considering an ‘increase’ following generic entry as meaning ‘an increase in level with no decrease in the trend OR an increase in trend with no decrease in level’ and taking a timepoint of 12 months following entry to assess the level change, 8/13 antibiotics exhibit an increase (Axtreonam, Cefpodoxime, Ciprofloxacin, Levofloxacin, Ofloxacin, Cefprozil, Cefuroxime axetil and Clarithromycin).

Now consider these results in the context of the issue you raised on p. 13 regarding research suggesting that “the biggest price reduction occurs after the second generic entry” (lines 277-278), and (lines 279-280) that “...for eight of the 13 antibiotics in our study, three or more generic products were approved one year after first generic entry (Appendix Table S1”. Now looking at Table S1 and the 8 antibiotics that had 3 or more generics 1 year after generic entry, 6 of these antibiotics (Cefprozil, Cefuroxime axetil, Ciprofloxacin, Clarithromycin, Levofloxacin and Ofloxacin) increased following generic entry according to the criteria I suggest above.

With all this in mind, wouldn't it be fairer to conclude that your results provide at least tentative evidence to suggest that there tends to be an overall positive effect of generic entry on antibiotic consumption in the US?

Response: Thank you for your comment and suggestion. We have now made changes to acknowledge both level and trend changes in our results and discussion. However, we note that the trend changes did not lead to consistent changes in level over time for cefprozil, cefuroxime axetil and clarithromycin, which was the case for the other antibiotics, and would be expected given the nature of the intervention. This could mean that the long-term impact is not the same for these three antibiotics (cefprozil, cefuroxime axetil and clarithromycin), and in worst case, that the change was caused by events occurring after the generic entry. We also note that the changes in cefprozil, cefuroxime axetil and clarithromycin (as well as aztreonam, cefpodoxime, and ofloxacin) represent use that levels out and stabilise. Because of this, we are cautious in our interpretation of these findings, and believe the most accurate way of describing the results is that there is no consistent pattern for antibiotic use following generic entry in the United States. See changes on page 2, lines 42-44, page 8, lines 202-204, page 9, lines 209-210, page 15, lines 318-320.

Reviewer comment: 3. I have some queries and comments on the calculations on absolute and relative changes in prescriptions (which I agree are very valuable to look at, including at the later timepoint of 24 months).

Firstly, I am a bit confused on how the calculations have been performed in places. For example, in the case of Levofloxacin, the change in level at 12 months was 672.41/m. We are told in line 178 (p. 8) that this is a relative change of 29%. This suggests that the base level was 2318.66/m. But in that case, surely the 1989.14/m increase at 24 months is an 86% of increase on this base, not a 125% increase as stated in line 190?

More broadly, in line with my concerns in point 2, when analysing the changes after 12 and 24 months, I think you should also consider the total changes, based on both the change in the level and

the change in the trend (i.e. compare the size of the increases against the projected values at these time points based on the pre generic entry trends).

Response: We have made additions in the Method and Material section explaining how we calculate relative change, page 7 lines 161-165.

Reviewer comment: Minor points to consider:

1. Please consider adding a newer reference to supplement references 3-4, i.e. Chatterjee A, Modarai M, Naylor NR, et al. Quantifying drivers of antibiotic resistance in humans: a systematic review. *Lancet Infect Dis.* 2018;18:e368–78.

Response: Thank you for your suggestion. The reference has been updated, page 3 line 57.

Reviewer comment: 2. Please explain/define all acronyms before using (E.g. on p.5, NDA/ANDA; on p. 6, IV)

Response: Thank you for bringing our attention to this. Acronyms have now been defined, page 5 lines 118-119, and page 8 line 187.

Reviewer comment: 3. On page 6, lines 142-143 under the Descriptive analysis subsection, we read “Nine of the 13 antibiotics had a declining trend prior to generic entry (Table 1),...” Maybe this is a bit pedantic, but the trends in the table are from a model (except possibly for Levofloxacin) so perhaps this shouldn’t go in this subsection. As an aside, is the declining trend for Levofloxacin from inspection of the inverted U-shape in Fig 1 or from a model?

Response: We agree, and have moved this section to “Overview of results”, see page 8 lines 204-207. We have also clarified that since the model for levofloxacin doesn’t generate a pre-intervention trend, the statement that use is declining for levofloxacin is based on visual inspection.

Reviewer comment: 4. On p. 8, line 181 – I think you mean ‘five’ rather than ‘six’ antibiotics here.

Response: We have changed the sentence to state “increase/decrease” since it refers to the six antibiotics that experienced significant level changes. See page 11 line 246.

Reviewer comment: 5. Identifying all other events (e.g. co-interventions) that may have influenced the results is of course infeasible/impossible. It is good that the study made some attempt to consider at least some major possible confounders. I would only suggest that, for full transparency and to give the reader more comfort, it would be good to provide a bit more detail on the methods you used, such as the search terms in PubMed, Google Scholar etc.

Response: We agree that this needs to be clarified and have made additions to the method section to elaborate on what was done to address co-interventions. We have also looked at some additional co-interventions, as suggested by the other reviewers, which has been added to the result section and discussion. Finally, we have clarified that the search was systematic

and conducted after we finished the ITS analyses, see pages 7, lines 166-182 (method and material), pages 12-13, lines 257-308 (results), pages 16-17, lines 347-372 (discussion).

Reviewer comment: 6. Perhaps include the sensitivity analysis described on p. 9 lines 195-197 in supplementary materials.

Response: We have now added the results from the sensitivity analysis in the supplementary material, Table S5 and S6.

Reviewer comment: 7. It is perhaps worth being clearer on explaining which direction the authors feel the market entry of Caystonem is more likely to have affected volume of Aztreonam (or stating that either direction seems plausible).

Response: We have made additions to clarify this on page 12 lines 273-278.

Reviewer comment: 8. In the Discussion, on p. 11 line 230 – perhaps I would weaken the statement that “...the increase in ciprofloxacin was partly the result of the anthrax scare...” to something like “...the results suggest that the increase in ciprofloxacin was partly the result of the anthrax scare...” as I don’t think we can be sure about this.

Response: Thank you for your suggestion, we have added this change in our manuscript, page 15 line 323.

Reviewer comment: 9. On p. 11 line 237, why is the fact that an antibiotic’s use was declining before generic entry a sign that lack of access is not an issue?

Response: We agree that this statement might be too speculative and have decided to remove it from the manuscript, page 15.

Reviewer comment: 10. Related to major point 2: on p. 12 line 259, it feels misleading to highlight that there only two cases of an upward trend after generic entry – at least without also mentioning that the change in the trend is positive in 7 cases.

Response: We believe that in order to understand our findings it is important to not only look at the results but also the type of change. A declining trend suddenly turning upwards signals a major impact on antibiotic use, compared to a change that could have been a natural levelling out since use cannot become “negative”, which we address in the discussion. However, we understand that the sentence can be considered somewhat misleading and have removed it.

Reviewer comment: 11. On p. 13, lines 273-275 it is claimed that the study’s approach “rules out the possibility that simultaneous generic entry of other antibiotics affected the antibiotics under consideration.” Surely this isn’t really true as each antibiotic was modelled separately in the study?

Response: We agree that even if there does not seem to be any interaction between the antibiotics included in the study, based on the graphs and time points of generic entry, we can’t rule out that this is truly the case. Therefore, we have decided to remove it from the manuscript, pages 17-18.

REVIEWER COMMENTS

Reviewer #2 (Remarks to the Author):

I appreciate the revisions made to the manuscript by Dr. Kallberg and colleagues. My original concern that the manuscript neglected secular trends has been mostly addressed. Additionally, I feel the methods and results are presented much more clearly in this revision. I have some additional comments that I think can easily be addressed for the authors' consideration.

- 1) Introduction: Reference #5 (the source of the statement that 30% of antibiotic use is unnecessary) is still not the primary source. I recommend you look at the 2016 manuscript by Fleming-Dutra et al. published in JAMA for this information.
- 2) Introduction: Minor comment: a study on prescribing factors in Delhi was included (reference #7). Given the immense differences in the U.S. and Indian healthcare systems and prescription drug markets, I would recommend removing this citation or contextualizing it more in the text.
- 3) Methods: Please clarify why standard units were used as the unit of analysis rather than prescriptions. I can see that in many cases the relationship between # of prescriptions and # of standard units would be parallel, but would any differences be expected in using these two measures? For example, would changes in recommended dosages or course length affect SUs? Do you anticipate any changes to your analysis if you used the # of prescriptions instead?
- 4) Results: All figure axes should start at 0.
- 5) Results: I appreciate the inclusion of additional information on secular trends. However, consider moving the presentation of information on secular trends for which no additional sensitivity analyses were conducted to the discussion section (where you already have some information).
- 6) Results: You mention that there are no national stewardship programs, however, the US Centers for Disease Control and Prevention's Get Smart stewardship initiative has been in place for many years, including during the timeframe of this study. This does not change your overall conclusion that stewardship creates gradual change compared with the more timely changes evaluated in this study, but I think many readers specializing in infectious disease or pediatrics are familiar with Get Smart and would expect to see it included. As noted above, this may be more appropriate in the discussion.
- 7) Results: Results are presented as prescriptions per population, but the methods indicate that SUs were used, so I am confused about what the unit of analysis is. Please clarify in the manuscript.
- 8) Discussion: As you noted, there was a marked decrease in cefdinir relative to what would be expected. Since this is the only drug where a decrease was noted, readers will naturally be curious about why this might be. You have this in your discussion already, I would recommend pulling it into its own paragraph and adding a little more information, if possible.
- 9) Discussion: Lines 338-340, you mention the increase in FQs is surprising. I think in the previous paragraphs you have justified why this increase may be observed, so I would consider removing the word surprising and using instead something like "concerning" or "notable".
- 10) Discussion: Some of the generic introductions occurred fairly close together (within a year or so). Is it possible to hypothesize about any potential changes in trends related to multiple generic entries? For example (and this is only a hypothesis), could the decrease in cefdinir be somewhat related to the earlier generic availability of azithromycin?
- 11) Discussion: Very minor correction – there are small typos in lines 304 and 305 (safety and therefore are misspelled).
- 12) Additional analysis: I am not sure how reliably this data is available, but a datapoint that might help better contextualize your study is the average price of these drugs prior to and after generic introduction. If this hypothesis is that lower generic prices spur higher use, then it would help the reader better understand observed changes if they could see the price change for each drug in concert with changes in drug quantities. For some of the drugs where no change was observed, is it possible that there was an insufficient change in drug price? I think this could be satisfied with a table in the appendix rather than adding it to the main analysis.
- 13) References: Minor comment: please check the reference formatting for all your references, some references seem incomplete.
- 14) Table 1 and 2: Please add the information on units to the table itself and a footnote defining the test used to calculate the p-value to help clarify the table for readers who may look at the tables before reading the text in detail.

Reviewer #3 (Remarks to the Author):

The effect of generic entry on antibiotic prescriptions in the United States, 2000-2012: An interrupted time series study

Summary of Paper

Antibiotic resistance is a major threat to public health, driven in part by overconsumption of antibiotics. When patents expire, branded antibiotics lose market exclusivity and generics typically enter the market at lower prices. This could potentially increase consumption of these antibiotics. This paper aims to examine the effect of generic market entry on antibiotic consumption in the United States. The method adopted is to use IQVIA data to conduct an interrupted time series analysis, spanning 2000-2012, of the change in volume of prescriptions per month per million people for antibiotics for which at least one generic entered the market. 13 different antibiotics, from several classes, are analysed. The paper reports that, one year after generic entry, prescriptions increased for 5/13 antibiotics, decreased for 1/13 and that there was no significant change for 7/13 antibiotics. The authors conclude that, in the United States, there is no consistent pattern for antibiotic use following generic entry.

As I wrote in my initial report, I think this paper makes a potentially valuable contribution to the literature. For example, if strong evidence can be found that generic entry is unlikely to increase antibiotic consumption, this would be useful to know as it would suggest that the main result from generic entry is positive (lower health care costs). On the other hand, if we were to learn that generic entry appears to increase antibiotic consumption, this could spur important future research on whether the increased antibiotic consumption following generic entry is due to meeting previously unmet needs or to overconsumption.

The revised paper has improved in many respects. I have still just one main reservation (and a few very minor ones) that I think the authors should address.

Main point to consider:

1. I am still just a little uneasy over the interpretation of the results. The discussion of the results in the paper focuses mainly around the changes in the prescription levels in the 13 antibiotics after generic entry. If the main hypothesis you wished to test was whether there was a change in level, visible 6-12 months after entry of first generic (or 18-24 months as sensitivity analysis) this is fine. If so though, I think you should it clearer that this was the main hypothesis. If it was pre-specified, then say so. If it was not, then please give a little more rationale, ideally with references, to help justify why a change in level rather than trend is the most plausible mechanic via which generic entry would increase antibiotic prescriptions.

In addition, in the Discussion, I do think a caveat should be added (perhaps following lines 329-331) that, despite the main conclusion that "...generic entry has limited and inconsistent effects on antibiotic use in the United States, with no significant, sustained, increase in more than half of cases", that we cannot ignore the fact that there was a positive significant change in trend for 7 of the 13 antibiotics (and only one significant negative change in trend). And that, though trends levelling out as sales approach zero might partly explain this, we cannot rule out the possibility that introduction of generics may lead to a change in trend that leads to much increased prescribing over longer time horizons.

Minor points:

1. The formula in line 149 uses Y_t to denote relative change at time t . However Y_t has previously been used in line 127 to denote total number of prescriptions per capita per month at time t . Please change the notation.

2. On line 172, cefuroxime is described as a "parenteral" treatment to distinguish it from IV - I

think this should read "oral"

3. I spotted just a few typos:

- a) On line 129 "and time t " should read "at time t "
- b) On line 134 "Appendix Figure 1" should read "Appendix Figure S1"
- c) On line 136 "where" should read "were"

REVIEWER COMMENTS

Reviewer #2 (Remarks to the Author):

Reviewer comment: 1) Introduction: Reference #5 (the source of the statement that 30% of antibiotic use is unnecessary) is still not the primary source. I recommend you look at the 2016 manuscript by Fleming-Dutra et al. published in JAMA for this information.

Response: We have now changed the reference to Fleming-Dutra et al., page 3, line 60.

Reviewer comment: 2) Introduction: Minor comment: a study on prescribing factors in Delhi was included (reference #7). Given the immense differences in the U.S. and Indian healthcare systems and prescription drug markets, I would recommend removing this citation or contextualizing it more in the text.

Response: We have decided to remove the reference.

Reviewer comment: 3) Methods: Please clarify why standard units were used as the unit of analysis rather than prescriptions. I can see that in many cases the relationship between # of prescriptions and # of standard units would be parallel, but would any differences be expected in using these two measures? For example, would changes in recommended dosages or course length affect SUs? Do you anticipate any changes to your analysis if you used the # of prescriptions instead?

Response: Thank you for making us aware of this mistake. Unfortunately, the revisions made in the previous round (where we were asked to describe the dataset in more detail) led to the wrong information being added. This group has previously worked with a similar dataset, where SU was used to measure use. However, it is the information about the dataset from the first version of the manuscript that is correct, where we state that use is measured in number of prescriptions. We have now corrected the manuscript, see page 5, lines 128-131.

Reviewer comment: 4) Results: All figure axes should start at 0.

Response: We have now made changes to the figures. However, in two cases the y-axis starts at with a negative value since projected levels of use goes below zero. Pages 10-11.

Reviewer comment: 5) Results: I appreciate the inclusion of additional information on secular trends. However, consider moving the presentation of information on secular trends for which no additional sensitivity analyses were conducted to the discussion section (where you already have some information).

Response: In response to the suggestion, we have now moved parts of the result section and incorporated it into the discussion. Page 16, lines 329-333, 340-343, 348-353, page 17, lines 375-383, page 18, lines 384-387.

Reviewer comment: 6) Results: You mention that there are no national stewardship programs, however, the US Centers for Disease Control and Prevention's Get Smart stewardship initiative has been in place for many years, including during the timeframe of this study. This does not change your overall conclusion that stewardship creates gradual change compared with the more timely changes evaluated in this study, but I think many readers specializing in infectious disease or pediatrics are familiar with Get Smart and would expect to see it included. As noted above, this may be more appropriate in the discussion.

Response: We have now included the Get Smart program in our discussion about antibiotic stewardship programs, page 17, lines 171-174.

Reviewer comment: 7) Results: Results are presented as prescriptions per population, but the methods indicate that SUs were used, so I am confused about what the unit of analysis is. Please clarify in the manuscript.

Response: Please see the response to comment #3. Prescriptions per population was used to measure the effect of generic entry, not standard units. We are sorry for this confusion.

Reviewer comment: 8) Discussion: As you noted, there was a marked decrease in cefdinir relative to what would be expected. Since this is the only drug where a decrease was noted, readers will naturally be curious about why this might be. You have this in your discussion already, I would recommend pulling it into its own paragraph and adding a little more information, if possible.

Response: We have moved this information to its own paragraph, page 18-19, lines 408-413.

Reviewer comment: 9) Discussion: Lines 338-340, you mention the increase in FQs is surprising. I think in the previous paragraphs you have justified why this increase may be observed, so I would consider removing the word surprising and using instead something like “concerning” or “notable”.

Response: We have exchanged “surprising” to “notable”, page 19 line 435.

Reviewer comment: 10) Discussion: Some of the generic introductions occurred fairly close together (within a year or so). Is it possible to hypothesize about any potential changes in trends related to multiple generic entries? For example (and this is only a hypothesis), could the decrease in cefdinir be somewhat related the earlier generic availability of azithromycin?

Response: We have now addressed this issue in the discussion, page 18, lines 392-407.

Reviewer comment: 11) Discussion: Very minor correction – there are small typos in lines 304 and 305 (safety and therefore are misspelled).

Response: We have corrected the typos.

Reviewer comment: 12) Additional analysis: I am not sure how reliably this data is available, but a data point that might help better contextualize your study is the average price of these drugs prior to and after generic introduction. If this hypothesis is that lower generic prices spur higher use, then it would help the reader better understand observed changes if they could see the price change for each drug in concert with changes in drug quantities. For some of the drugs where no change was observed, is it possible that there was an insufficient change in drug price? I think this could be satisfied with a table in the appendix rather than adding it to the main analysis.

Response: We agree that looking at price changes for each individual product could add valuable context, but such data is not readily accessible, and we have therefore not been able to procure historic drug prices from providers of original and generic versions of the drugs. Average price for each antibiotic could be of interest. However, research show that introduction of generics can lead to an increase in the price of branded products. As a result, the decrease in drug price, following generic entry, is not always reflected in the average price (Schondelmeyer, Stephen W., and Leigh Purvis. *Trends in retail prices of brand name prescription drugs widely used by Medicare beneficiaries 2005 to 2009*. AARP, Public Policy Institute, 2010. and Aitken, Murray L., et al. "The regulation of prescription drug competition and market responses: patterns in prices and sales following loss of exclusivity." *Measuring and Modeling Health Care Costs*. University of Chicago Press, 2013. 243-271).

Reviewer comment: 13) References: Minor comment: please check the reference formatting for all your references, some references seem incomplete.

Response: We have now checked all references and made changes where needed.

Reviewer comment: 14) Table 1 and 2: Please add the information on units to the table itself and a footnote defining the test used to calculate the p-value to help clarify the table for readers who may look at the tables before reading the text in detail.

Response: We have now added this information to the tables.

Reviewer #3 (Remarks to the Author):

Main point to consider:

Reviewer comment: 1. I am still just a little uneasy over the interpretation of the results. The discussion of the results in the paper focuses mainly around the changes in the prescription levels in the 13 antibiotics after generic entry. If the main hypothesis you wished to test was whether there was a change in level, visible 6-12 months after entry of first generic (or 18-24 months as sensitivity analysis) this is fine. If so though, I think you should it clearer that this was the main hypothesis. If it was pre-specified, then say so. If it was not, then please give a little more rationale, ideally with references, to help justify why a change in level rather than trend is the most plausible mechanic via which generic entry would increase antibiotic prescriptions.

In addition, in the Discussion, I do think a caveat should be added (perhaps following lines 329-331) that, despite the main conclusion that "...generic entry has limited and inconsistent effects on antibiotic use in the United States, with no significant, sustained, increase in more than half of cases", that we cannot ignore the fact that there was a positive significant change in trend for 7 of the 13 antibiotics (and only one significant negative change in trend). And that, though trends levelling out as sales approach zero might partly explain this, we cannot rule out the possibility that introduction of generics may lead to a change in trend that leads to much increased prescribing over longer time horizons.

Response: Thank you for your comment. We have now revised the method section and discussion to address these issues. We have clarified that the pre-specified hypothesis was a level increase within 6-12 months, and by 24 months at the latest. Time points were chosen in accordance to research showing that it is reasonable to expect to see an effect of generic entry within this time period. Also, the same or similar time points have been used in other research analysing the effect of generic entry. We have also specified that, as a second hypothesis, we look at immediate change in trend, and have revised the discussion acknowledging the potential long-term effect of these changes. We have avoided describing these changes as "much increased" since this relates to discussions about the clinical relevance of changes in antibiotic use. This would require data linking levels of antibiotic use to levels of antibiotic resistance, and has not been discussed for any of the other antibiotics, pages 4 lines 105-106, page 5 lines 107-122, page 19, lines 424-427.

Minor points:

Reviewer comment: 1. The formula in line 149 uses Y_t to denote relative change at time t . However Y_t has previously been used in line 127 to denote total number of prescriptions per capita per month at time t . Please change the notation.

Response: We have exchanged Y to R , page 7, lines 177-178.

Reviewer comment: 2. On line 172, cefuroxime is described as a "parenteral" treatment to distinguish it from IV – I think this should read "oral"

Response: We have now changed it to oral, page 8, line 200.

Reviewer comment: 3. I spotted just a few typos:

- a) On line 129 “and time t” should read “at time t”
- b) On line 134 “Appendix Figure 1” should read “Appendix Figure S1”
- c) On line 136 “where” should read “were”

Response: Thank you for detecting the typos, they have now been corrected, page 6, line 156, page 7 lines 161 and 164.